# Self-Employed Canadians' Experiences with Cancer and Work: A Qualitative Study

Christine Maheu [1,*], Maureen Parkinson [2], Caitlin Wong [1], Fatima Yashmin [1] and Caroline Longpré [3]

1   Faculty of Medicine and Health Sciences, Ingram School of Nursing, McGill University, Montreal, QC H3A 2M7, Canada
2   BC Cancer Agency, Patient & Family Counseling, Vancouver, BC V5Z 1G1, Canada
3   Nursing Sciences Department, Université du Québec en Outaouais (UQO), St-Jérôme, QC J7Z 0B7, Canada
*   Correspondence: christine.maheu@mcgill.ca

**Abstract:** Self-employed individuals with cancer face unique challenges that have not been fully explored in previous research. For example, while some studies conducted in Europe have suggested that self-employed workers with cancer experience more adverse health and work-related outcomes compared to salaried workers, the specific manners in which cancer affects the health, work, and businesses of self-employed individuals remain inadequately understood. This lack of understanding represents a crucial gap in the literature, as self-employed individuals make up a significant portion of the workforce in many countries, including Canada. To address this gap, a qualitative interpretive description study was undertaken to explore the experiences of 23 self-employed Canadians diagnosed with cancer from six provinces, with the aim of generating insights into the unique challenges faced by this population. The interviews were conducted in the language chosen by the participants from the two official languages of Canada, namely English and French. Using reflexive thematic analysis, four major themes and twelve subthemes were generated from the participants' shared accounts that reflected the impact of cancer on self-employed Canadians' physical, cognitive, and psychological function, affecting their work ability and their ability to maintain their business and financial well-being. Participants in the study also shared strategies they used to continue working and maintain their business during their cancer experience. This study sheds light on the impact of cancer on self-employed individuals and provides insight into the experiences of self-employed individuals with cancer that can inform the development of interventions to support this population.

**Keywords:** self-employment; cancer; vocational rehabilitation; return to work; maintain work





## 1. Introduction

Reduced and impaired work ability can be one of the most challenging consequences of cancer and its treatment, resulting from the impact of cancer on physical, mental, and cognitive functions [1–3]. For self-employed individuals with cancer, the impact of cancer on physical, mental, and cognitive functions frequently leads to reduced and impaired work ability, which can be one of the most challenging consequences of cancer and its treatment [1,2,4]. Work ability refers to a person's physical, psychological, and social ability and resources for engaging in any type of paid work or self-employment and reflects a work outcome [2]. Declines in these clinical and work outcomes in individuals diagnosed with cancer are often associated with increased absenteeism and presenteeism, financial insecurity, job loss, and early retirement [5–7]. Self-employed individuals with cancer face additional adverse work outcomes with the risk of business closure, loss of their only source of income, possible loss of future business contracts, and harm to their business reputation [8–10]. The absence of support and resources available to self-employed workers, particularly in the context following a cancer diagnosis, can further exacerbate these negative work outcomes [11–13]. Few studies have investigated the impact of cancer on the

work and business of self-employed workers. Among the studies identified, primarily conducted in Europe, they found that self-employed workers diagnosed with cancer experience more adverse health and work-related outcomes than cancer-diagnosed salaried workers, despite taking fewer disease-related absences than their salaried counterparts [8,14]. While self-employed individuals with cancer are more likely to continue working during treatment [14], they do so with worse clinical, work, and financial outcomes [8,13].

In Canada, self-employment is an important and growing part of the workforce, with approximately 15% of workers being self-employed [15,16]. Recent Canadian cancer statistics indicate that a significant number of cancer diagnoses occur among working-age individuals aged 15 to 65 [17–19]. When surveyed, self-employed Canadians with cancer report experiencing reduced income that is 37% to 48% higher than that of employed Canadians with cancer [20]. Although there is an Employment Insurance (EI) benefits program for self-employed Canadians, few register for it, as the program is relatively expensive and requires meeting the minimum qualifying period, proving a substantial loss of working time, and being subject to a payment limit that may not cover their basic living expenses, in addition to a 26-week time limit. Although there is increasing interest in understanding the impact of cancer on the work and business of self-employed individuals, few studies have investigated the distinctive circumstances and obstacles faced by self-employed Canadians with cancer.

This qualitative interpretive description study aimed to explore the impact of cancer on the health, work ability, and business of self-employed Canadians. It also aimed to identify the challenges they faced and the strategies they used to continue working and maintaining their business while managing their cancer. Finally, the study sought to suggest potential support programs or policies that could better meet the needs of this population.

## 2. Materials and Methods

### 2.1. Study Design

This study employed a qualitative interpretive description design, which was chosen for its approach to derive rich, contextually specific insights, patterns, and themes from complex and dynamic healthcare phenomena that can inform healthcare practices, policies, and education [21–24].

In total, 23 self-employed Canadians with cancer participated in this study, taking part in in-depth interviews that utilized open-ended questions to explore the impact of cancer on their health, work ability, and business. The interview questions were initially guided by the vocational rehabilitation (VR) model for cancer survivors [25], which provides a framework for understanding the experiences of returning and maintaining work after cancer. Reflexive thematic analysis was used to analyze the data, which facilitated the identification of patterns and themes [26,27]. The study followed standards and completed the checklist for reporting qualitative study as outlined in the "COREQ" (consolidated criteria for reporting qualitative research) guidelines (File S1) [28]. Excel® was used to manage and analyze the data.

### 2.2. Participant Selection and Method of Approach

Using a combination of convenience and snowball sampling techniques, this study recruited eligible participants who were bilingual in either English or French, the two official languages of Canada, from across the country. This sampling approach is a rare example of bilingual recruitment for studies of this nature in Canada. Participants were recruited through various means, including posting advertisements at major cancer centers in three urban cities—Montréal, Québec, and Vancouver—as well as through the Canadian Cancer and Work website (www.cancerandwork.ca, accessed on 24 April 2023), owned by the first and second authors (C.M. and M.P.), and the Cancer and Work LinkedIn account. Additionally, enrolled participants were asked to identify and communicate with other self-employed individuals who they knew had been diagnosed with cancer and

might be interested in participating (snowballing). This last strategy identified 50% of the study participants.

A total of 23 eligible participants were included in the study, meeting the following criteria: (i) being a self-employed Canadian who owns a business or works as a freelance consultant; (ii) having received a cancer diagnosis while self-employed; (iii) being at least 18 years old, with no upper age limit as long as they were still working as a self-employed at the time of their diagnosis; and (iv) speaking either English or French. Although having access to a computer with internet access for conducting the interview online was preferred, participants were given the option to interview over the telephone, if necessary, but it was not mandatory. No eligible participants declined to participate once approached or after consenting to take part in the study.

All potential participants were invited to participate in the study through a personalized link to Qualtrics, a secure web-based electronic data collection platform. An electronic consent form and a demographic survey were sent along with the invitation. The academic institution of the first author has a license with Qualtrics to ensure the privacy and security of the research data. Participants who agreed to participate were required to check the consent box on the electronic form and complete the demographic, clinical, and work-specific questionnaires.

After consenting and completing the questionnaires, the research team contacted the participants to schedule a time and date for the interviews. Due to the dispersed nature of the target population, the interviews were conducted using virtual platforms. For participants without internet access, an option to conduct the interview over the phone was offered. However, the call was initiated through the Zoom® platform to take advantage of its benefits. Zoom® was chosen as the interview platform because the first author (C.M.) obtained a faculty license from their academic institution. The platform's secure recording feature facilitated data management and security, and real-time transcription in multiple languages was also available.

*2.3. Data Collection*

Between 2021 and 2022, 23 self-employed Canadians participated in semi-structured interviews that lasted from 45 to 90 min. The interviews were conducted by the first author (C.M.) and one other co-author (F.Y., C.W., or M.P.), depending on their availability. C.M. and M.P. have extensive experience in cancer and work, while C.W. and F.Y. were graduate nursing students in training under the supervision of C.M. To ensure transparency and facilitate discussion, the study interview guide was shared with the participants at least two days prior to the interview. The interview guide was reviewed for relevance and clarity by three self-employed individuals with cancer before the interviews began, further ensuring transparency and comprehensibility.

The interview guide for this study was structured around the four factors of the cancer survivor VR model, which are known to have implications for cancer survivors' vocational outcomes [25]. However, to ensure that the interviews remained open to meaningful discussions, the guide was loose and flexible enough to allow any other relevant topics to emerge. As a result, the discussions covered a range of topics related to cancer and self-employment while ensuring that all four main areas of the VR model were addressed. An open and flexible approach to interviewing that remains modifiable and transformative as themes evolve during analysis [29] is recommended for qualitative interpretive descriptive studies that employ reflexive thematic analysis to analyze and interpret their qualitative data [26,27]. The interview questions covered topics such as the impact of cancer on participants' health, work ability, and businesses; changes in their views on work since their cancer diagnosis; challenges in maintaining their work and businesses; strategies used to manage work and cancer; and support programs or policies that could assist self-employed Canadians with cancer. Participants were also invited to share any additional experiences related to self-employment and cancer.

During the data collection phase of this study, various measures were taken to ensure that the interviews were conducted in a thorough and transparent manner. Repeat interviews were not carried out, but audio recordings and field notes were made during and after each interview. The duration of the interviews varied between participants, with a range of 45 to 90 min. Data saturation was discussed during the analysis phase to ensure that the research objectives were met, and that sufficient data had been collected. To further ensure transparency and accuracy, the interview transcripts were shared with three self-employed individuals with cancer for comment and correction. This collaborative process helped validate the data and ensure the participants' perspectives were accurately captured in the final analysis.

*2.4. Data Analysis*

A primarily inductive, data-driven "open-coded" reflexive thematic analysis approach [27] was used to derive and generate patterns and themes meaningful to the participants' data-based accounts and relevant to the research questions [26]. Initially, coding was performed using the "comments" feature in Microsoft Word, following Byrne's [26] worked example of Braun and Clarke's approach to reflexive thematic analysis. Afterward, all codes and corresponding interview segments were transferred to an online Excel spreadsheet accessible to the research team to facilitate iterative analysis and the formulation of definitions for the generated themes and sub-themes. The six-phase analytical process proposed by Braun and Clarke was adopted for the iterative analysis of the study dataset.

During the familiarization phase (Phase 1), the first author, C.M., and one of the two graduate nursing students (C.W. and F.Y.) conducted a close reading of the dataset to identify relevant information and initial observations, using "in vivo" coding to generate initial codes (Phase 2), and capturing sufficient detail from the participants' accounts to be able to stand alone and inform the underlying commonalities among the data items [26]. During the initial "in vivo" coding process, codes were captured using concise and descriptive labels relevant to the research questions. Some codes, such as "putting my professional license at risk with my difficulty when needing to think" and "having job fatigue," were grouped together under the "Impact of cancer on health" category. Other categories were also created to group the codes, which were further organized under sub-themes contributing to the candidate theme, which in the example above became "Impact of cancer on self-employed individuals' function". The expanding list of codes was periodically reviewed by the team. M.P.'s (co-author) extensive personal and professional experience with cancer and vocational rehabilitation provided valuable insights and feedback on the codes generated during the study, ensuring that the codes accurately captured the complex phenomena studied and were meaningful and relevant to the research questions.

In the theme-generating phase (Phase 3), the team engaged in a collaborative discussion of all the codes and categories, working towards developing overarching themes and sub-themes that captured the dataset's overall significance. During the recursive review phase (Phase 4), the team conducted a detailed analysis of the sub-themes and candidate themes, assessing their quality, determining their boundaries, and ensuring sufficient meaningful data supported them. This phase required restructuring some of the candidate themes and sub-themes, as well as adding and removing codes to produce a coherent and comprehensive thematic framework. Following, the team engaged in analyzing and providing in-depth interpretations of how each theme and sub-theme related both to the dataset and to the research questions, revising and renaming the themes, if necessary, and selecting the data extracts that were to support the accounts made by each theme to arrive at an illustrative thematic framework, contextualizing the sub-themes within their respective themes.

In sum, after identifying codes that best represented the participants' stories while remaining true to their own words, the team categorized the codes into possible sub-themes and candidate themes, allowing them to identify patterns in the data and draw conclusions about the participants' experiences and perspectives. The team then further analyzed the

sub-themes by linking them to the VR model for cancer survivors, which encompasses four factors: biopsychological, person-related, system-related, and worksite-related [25]. If a sub-theme related to one or more of these factors, the corresponding VR factor was identified in brackets beside the sub-theme. The four factors of the VR model for cancer survivors can be barriers and facilitators to returning to and maintaining work following cancer. Biopsychological factors include treatment side effects, previous co-morbid conditions, and mental health issues. Person-related factors relate to personal views on work–life balance and the significance of work, as well as sociodemographic factors such as age and education, which can influence remaining at or returning to work. System-related factors represent the support systems surrounding the individual, such as healthcare, rehabilitation, financial resources, family, insurance, and legal resources. Worksite factors relate to the job context, job demand characteristics (cognitive, physical, and mental), type of work and work hours, support at work (job accommodations and supervisor/coworker support), and workplace relationships. Assessing the work outcome experiences of individuals with cancer within these factors can help to identify facilitators and barriers to returning to work, explain work-related challenges that individuals with cancer may face, and provide a guide for identifying areas for support in the return to work process [25]. Table 1 presents examples of how the participants' raw data, presented as quotations, were analyzed and linked to specific themes and factors in the VR model. To ensure traceability and identification of each participant's quotations, a unique participant number was assigned to each quote and used throughout the analysis process.

**Table 1.** Examples of the analysis of participants' data and linkage to the study themes and VR model factors.

| Quotes from Participants | Codes | Sub-Themes | Candidate Themes |
|---|---|---|---|
| "I felt physically drained, but I had no choice but to return to work two weeks after my last chemotherapy treatment. I needed to earn a living, but more importantly, my clients needed me. I was absolutely exhausted" (P4). | Having to work despite physical exhaustion | Impact of cancer on physical function (VR factor: biopsychological) Impact of cancer on their financial well-being | Impact of cancer on self-employed individuals' function Impact of cancer on the self-employed financial well-being |
| "When it came to big management decisions, there was nobody else" (P13). | Inability to find suitable replacements | Impact of cancer when have key positions in organization (VR factor: person-related and worksite) | Impact of cancer on self-employed ability to maintain their business |
| "Not having a job or unemployment insurance meant that if you didn't work, you had no income" (P22). | Financial vulnerability brought upon cancer diagnosis | Struggling to maintain a stable income during cancer (VR factor: person-related, system, and worksite) | Impact of cancer on self-employed financial well-being |
| "When they did ask healthcare providers for advice, seven participants said their treatment centers were accommodating, allowing them to come in for treatment when they could and when it was best for them" (P12). | Flexible healthcare scheduling of cancer treatment | Strategies for maintaining and/or returning to work (VR factor: system) | Facilitating factors for working with cancer |

### 2.5. Ethical Considerations

The Institutional Review Board of McGill University Health Centre (MUHC) granted the primary correspondence author (C.M.) ethical approval for the study (MM-2021-6745). Each participant signed a consent form in their interview language. A code number was assigned to each participant to protect their anonymity.

## 3. Results

In total, 23 self-employed Canadians with cancer participated in this study, representing six provinces. Of the 23 participants, 19 spoke English, while 4 spoke French. The age range of participants was broad, though over half were aged 40–59. The most commonly diagnosed cancer type was breast cancer, which affected 30% of participants. Of the 23 participants, 14 identified as female and nine as male. Participants included storefront owners, family business owners, service providers (e.g., floor and towing companies, accountants, and primary care providers), and independent contractors (e.g., contract writers), with some businesses having a physical storefront and others operating online. Table 2 provides a comprehensive overview of the participants' demographic, clinical, and work-related characteristics and work status following their cancer diagnosis using the Canadian National Occupational Classification (NOC) system [30].

**Table 2.** Characteristics of the 23 self-employed Canadians with cancer.

| Age at Time of Cancer Diagnosis | N | % |
|---|---|---|
| 20–39 | 4 | 18% |
| 40–59 | 12 | 52% |
| 60+ | 7 | 30% |
| **Highest Level of Education** | | |
| Did not complete high school | 1 | 4% |
| CEGEP/college/technical program | 6 | 26% |
| University | 16 | 70% |
| **Type of Cancer** | | |
| Breast | 7 | 30% |
| Prostate | 6 | 26% |
| Thyroid | 3 | 13% |
| Lymphoma/Leukemia | 2 | 9% |
| N of 1 for Other: colorectal, kidney, bladder, brain, stomach | 5 | 22% |
| **Province of Canada** | | |
| British Columbia | 3 | 13% |
| Alberta | 1 | 4% |
| Saskatchewan | 1 | 4% |
| Manitoba | 3 | 13% |
| Ontario | 9 | 40% |
| Quebec | 6 | 26% |
| **Year of Initial Cancer Diagnosis** | | |
| 2015+ | 13 | 57% |
| 2005–2014 | 9 | 39% |
| Unanswered | 1 | 4% |
| **Years Self-Employed** | | |
| >5 years | 4 | 17% |
| 5–14 years | 5 | 22% |
| 15+ years | 10 | 43% |
| Unanswered | 4 | 17% |
| **Job Classification (NOC)** | | |
| Business, finance, administrative management | 2 | 9% |
| Health occupation | 4 | 17% |
| Education, law, social, community, and government services | 3 | 13% |
| Art, culture, recreation, and sport | 1 | 4% |
| Manufacturing and utilities | 1 | 4% |
| Sales and service occupations | 9 | 40% |
| Trades, transport, and equipment operators and related occupations | 2 | 9% |
| Retired (immediately following cancer diagnosis) | 1 | 4% |

**Table 2.** *Cont.*

| Work Status after Cancer Diagnosis | Participant ID |
|---|---|
| Remained working following diagnosis and took less than the equivalent of 4 weeks off work during the active treatment phase | 5, 9, 11, 12, 13, 19, 21, 22 (*n* = 8) |
| Initially attempted to continue working, but ultimately closed the business due to cancer's impact on their ability to perform essential tasks | 5, 16, 20, 23 (*n* = 4) |
| Contracted out services/reduced their business role/reduced client or contract load | 3, 9, 11, 13, 14, 17, 18, 19, 22 (*n* = 9) |
| Changed to salaried work | 7, 15, 23 (*n* = 3) |
| Retired early | 3, 12, 14 (*n* = 3) |
| Temporarily closed the business for more than 1 month to at least 12 months | 1, 2, 3, 4, 6, 7, 8, 10, 15, 16, 17, 20, 23 (*n* = 13) |

Through a thematic and constant comparative analysis of the transcribed interviews, four major themes and twelve sub-themes describing the impact of cancer on self-employed Canadians' work and business, as well as strategies they used to remain at work with cancer or return to work after cancer, were identified (see Figure 1: Navigating Cancer and Work: Thematic Insights from Self-Employed Individuals). The four themes include (1) the impact of cancer on self-employed individuals' function, (2) their ability to maintain self-employment, (3) their financial well-being, and (4) self-employment factors that facilitate working with cancer.

**Theme 1. Impact of cancer on self-employed individuals' function**

1.1 Impact of cancer on self-employed physical function (Biopsychological VR Factor)
1.2 Impact of cancer on self-employed cognitive function (Biopsychological VR Factor)
1.3 Impact of cancer on self-employed psychological function (Biopsychological VR Factor)

**Theme 2. Impact of Cancer on Self-Employed Ability to Maintain their Business**

2.1 Impact of cancer on key position holders in the organization (Person-Related, and Worksite VR Factors)
2.2 Keeping the business running for the clients (Person-Related, and Worksite VR Factors)
2.3 Maintaining the business in operation for their employees (Person-Related, and Worksite VR Factors)
2.4 Their business is their legacy and their social network (Person-Related VR Factor)

**Theme 3. Impact of Cancer on Self-Employed Financial Well-Being**

3.1 Struggling to maintain a stable income during cancer (Person-Related, System-Related, and Worksite VR Factors)
3.2 Financial impact of losing their gamble to cancer (System-Related VR Factor)

**Theme 4. Facilitating Factors for Working with Cancer**

4.1 Ability of the self-employed to manage their work schedule, workload, and tasks to be completed (Worksite VR factor)
4.2 Support from the healthcare team and the worksite (System-Related and Worksite VR Factors)
4.3 Strategies for maintaining and/or returning to work (System-Related VR factor)

**Figure 1.** Navigating Cancer and Work: Thematic Insights from Self-Employed Individuals.

### 3.1. Theme 1: Impact of Cancer on Self-Employed Individuals' Function

Theme 1 describes the impact of cancer and its treatment side effects on the physical, psychosocial, and cognitive functions of self-employed individuals and how changes in these functions affect work ability. Participants reported feeling ill-prepared for how cancer could potentially impact their work ability (physical and mental) and their ability to maintain their business operations. For example, one participant stated, "you don't have time to adjust weeks after diagnosis you're supposed to be balancing with work, it's pretty hard" (P2). When first diagnosed with cancer, none of the 23 participants immediately stopped working. Rather, to manage continuing to work and the treatment side effects, participants asked their healthcare providers if they could schedule their treatment on days preceding a period in which they could be away from work. Working from home was also

noted as helpful in managing treatment side effects. One participant, whose work primarily involved physical labor, declined a recommended treatment to avoid potential side effects that could have disrupted their work and compromised their physical work ability. All participants experienced a change in employment status during their cancer trajectory to the recovery phase, with some reducing work hours or eventually stopping work due to decreased work function capacity.

### 3.1.1. Impact of Cancer on Self-Employed Physical Function (Biopsychological VR Factor)

Eight participants in the study reported that cancer and its treatment had a significant impact on their physical functioning, affecting their ability to meet the physical demands of their jobs. Lymphedema and fatigue were common concerns that affected work ability. For example, one participant developed lymphedema in the arm, making it impossible to wear an arm sleeve while performing job tasks that required hand use. Wearing the arm sleeve made it difficult for the participant to perform the work correctly, ultimately leading to the temporary closure of the business.

Participants responded differently to the physical limitations caused by cancer and its treatment. Some chose to continue working despite feeling exhausted, as exemplified by this quote: "I had no choice but to return to work two weeks after my last chemotherapy treatment. I needed to earn a living, but more importantly, my clients needed me. I was absolutely exhausted" (P4). Others delegated the more strenuous work to employees or contractors (P6, P12, P18, P17) when possible. Thirteen of the twenty-three self-employed participants in the study were forced to temporarily close their businesses for at least one month and up to one year due to the impact of cancer on their decreased physical ability to perform essential job duties. This resulted in significant financial losses, including decreased income, depleted savings, reduced pensions, and financial worries. Additionally, these participants described being unable to meet the physical demands of their jobs or make additional adjustments to continue working.

### 3.1.2. Impact of Cancer on Self-Employed Cognitive Function (Biopsychological VR Factor)

Participants reported experiencing cognitive changes that affected their mental work capacity, including impaired concentration and memory. These changes were of particular concern for those in licensed professions who feared risking their license. For instance, one participant shared, "Although I had no memory of executing decisions, my signature would appear on documents" (P7). To cope with these changes, some wrote things down or delegated mentally demanding tasks. Participants also reduced their work hours and client interactions, which posed a risk to their business viability. One participant had planned to work while undergoing cancer treatment, but their business closed due to the pandemic. This closure was a "blessing" for their recovery, as cancer treatment had left them physically, emotionally, and mentally drained. They noted that without the government's pandemic-related financial aid, they would not have been able to afford to take sick leave for cancer treatment because their business was their sole income source.

### 3.1.3. Impact of Cancer on Self-Employed Psychological Function (Biopsychological VR Factor)

"When I first received my diagnosis, it had a significant impact on my ability to work, not physically but emotionally. I just never imagined I would be diagnosed with cancer. My sleep was also affected, and I found myself sleeping until noon or later because I couldn't sleep well at night. I would get up and go through the motions of getting ready, but I wouldn't bother with makeup or fixing my hair, and then I never made it to the office. I would just sit at home. That was the extent of what I could handle at that time" (P1).

This passage highlights the emotional impact of cancer on self-employed individuals, and the verbatim quote from Participant 1 adds a personal and specific view of the possible

impact of cancer on self-employed psychological function. For further insight into the impact of cancer on psychological function and its barrier to returning and maintaining work, refer to the testimonial video featuring Frédéric's experience (see Video S1: Testimonial from Frédéric, available at www.mdpi.com/xxx/s1).To further describe how cancer can negatively impact self-employed psychological function, affecting their work ability, participants reported experiencing anxiety, fear, and distress in response to their initial cancer diagnosis. The cancer diagnosis was experienced as a shock, and the emotional reactions varied greatly. Participants reported being unable to afford a "meltdown" and felt compelled to maintain business operations in order to maintain income and employee employment. They kept their "worry time" to a minimum. Two participants believed that their cancer experiences enhanced their ability to connect emotionally with their clients. In contrast, two other participants (P15, P10) reported losing all passion and emotional attachment to their company, which had a devastating effect on their work productivity. Three participants experienced depressive episodes that hindered their ability to work, and one felt stuck in this emotional state for eight years after their diagnosis. These participants feared they could no longer provide their clients with the same level of quality work and were afraid of making mistakes at work because the cancer experience had emotionally drained them.

In brief, the study revealed that self-employed individuals with cancer faced significant challenges in maintaining their businesses due to the impact of cancer on their physical, cognitive, and psychological/emotional function, affecting their work ability.

### 3.2. Theme 2: Impact of Cancer on Self-Employed Ability to Maintain Their Business

This theme encompasses four sub-themes that illustrate the self-employed participants' deep connection to their businesses. The participants saw their businesses as a crucial part of their legacy, and the sub-themes demonstrate how their personal and professional lives often intersected. This was especially true for those who owned family businesses, which furthered their sense of duty to keep working.

#### 3.2.1. Impact of Cancer on Key Position Holders in the Organization (Person-Related and Worksite VR Factors)

Participants who held key positions in their organizations continued working during cancer treatment out of devotion to their business, employees, and clients, and to maintain their income. Some participants delegated some day-to-day tasks to others while still making significant management decisions themselves since there was no one else in the business who could do so (P13). Others used technology to continue managing their work and staff from their hospital beds.

#### 3.2.2. Keeping the Business Running for Clients (Person-Related and Worksite VR Factors)

Participants' initial response to receiving a cancer diagnosis was to maintain service for themselves, their employees (if any), and their clients, even if they did not provide the services directly. One participant noted, "If you don't provide a service, you don't have a business" (P12), while another observed that "you just can't expect your business to grow during this period" (P17). Participants who were sole proprietors and had no other source of income would suffer significant financial losses if they could not work during treatment. They also feared losing customers if they had to interrupt services. Some participants used their savings to hire a contract consultant to perform some of the work while they were temporarily unable to fulfill their business duties due to cancer.

#### 3.2.3. Maintaining the Business in Operation for Their Employees (Person-Related, and Worksite VR Factors)

Ten out of the twenty-three participants who had employees expressed a sense of obligation to maintain their business operations to ensure the continued employment of their staff. Despite believing that their businesses could have survived a temporary pause, some business owners continued their operations out of loyalty to their employees. Others

worked part-time to secure contracts for their staff, given their own struggles with reduced physical, mental, and cognitive stamina resulting from cancer and its treatment. In keeping the business running, employees provided invaluable support by taking on additional responsibilities such as heavy lifting, bookkeeping, and routine tasks. The participants cited the support and understanding of their employees as a significant factor that helped them sustain their businesses throughout their cancer journey.

### 3.2.4. Their Business Is Their Legacy and Their Social Network (Person-Related VR Factor)

For many of the participants, their work was their life. As P2 explained, "I never imagined working for someone else. Being my own boss and having the freedom to align my work with my personal values and express my creativity without restrictions was incredibly important to me". They viewed their businesses as an integral part of their legacy and contribution to society, and they were committed to preserving them despite the difficulties of managing their cancer at the same time. Family business owners, in particular, felt a strong sense of obligation to continue working, given the close intertwining of their personal and professional lives. Clients who shared the participants' values became an important part of their social and support networks. While some participants could not maintain their businesses during the active state of their cancer, they remained committed to expanding their legacy and their business, highlighting the strong connection between their personal and professional lives.

### 3.3. Theme 3: Impact of Cancer on Self-Employed Financial Well-Being

Theme 3 focuses on the difficulties self-employed individuals face in maintaining their financial stability while going through cancer. Numerous participants struggled to maintain a stable income during cancer, prompting them to seek alternative sources of financial support. This included exhausting personal savings, requesting assistance from family members, obtaining bank loans, and turning to external sources.

### 3.3.1. Struggling to Maintain a Stable Income during Cancer (Person-Related, System-Related, and Worksite-Related VR Factors)

When cancer interfered with the ability of self-employed individuals to maintain their business operations and income, they were required to seek alternative sources of financial support. This resulted in the need to seek alternative sources of financial support, such as exhausting personal savings, borrowing money, or taking out bank loans. Financially vulnerable participants found it difficult to maintain a steady income, and 11 relied on their spouses for support. Eleven participants reported that their financial situation remained stable throughout their cancer treatment, but they continued to work to earn a living and keep their businesses afloat. As one participant (P22) explained, "Not having a job or unemployment insurance meant that if you didn't work, you had no income". Three participants had to transition to salaried employment with less demanding job requirements to ensure a stable income after experiencing financial difficulties as self-employed individuals with cancer. During the pandemic, some participants received government relief funding, which they viewed as a "blessing"; however, they were required to plan their finances carefully. One participant utilized the GoFundMe platform to create a fundraising page for herself, as she had no other means of financial support.

### 3.3.2. Financial Impact of Losing Their Gamble to Cancer (System-Related VR Factor)

Several participants faced financial difficulties as a result of their cancer, resulting in decreased work and business productivity and increased expenses. Although some participants were aware of the employment insurance (EI) benefits offered by the Canadian government to self-employed individuals, more than half were unaware of the program. Participants reported choosing to forgo EI coverage due to the high cost of premiums, with only one participant opting for private EI coverage. Instead, they took a financial risk by "gambling" and relying on their savings in case of illness, hoping they would not need

to take time away from work, "I chose to take a chance and never need unemployment insurance" (P22). However, participants recognized the importance of private or public insurance in ensuring financial stability during treatment. The financial impact of cancer also led to additional expenses such as travel and lodging for treatment, loss of contracts, and reduced earning capacity. In order to reduce expenses, some participants sold their homes or moved in with relatives.

### 3.4. Theme 4: Facilitating Factors for Working with Cancer

Factors that facilitated working with cancer included the characteristics of the job, the type of business operated, and the support obtained from external systems such as healthcare teams, family, and the work site. Strategies used by self-employed individuals with cancer to maintain work during and after their cancer treatment were also helpful.

#### 3.4.1. Ability of the Self-Employed to Manage Their Work Schedule, Workload, and Tasks to Be Completed (Worksite VR Factor)

Self-employed individuals who had sufficient control over their work to make executive decisions regarding their work schedule, workload, and tasks to be completed had an easier time working with cancer. Fourteen of the twenty-three participants fell under this category and were able to modify their work schedule, workload, and tasks to accommodate their treatment timeline and their fluctuating cancer-related functional impairments. Nevertheless, even with this control, these alterations still necessitated advance planning to minimize disruptions to the workplace. Participants who owned businesses that employed workers found it easier to implement these modifications to maintain or resume their work. Ultimately, participants attempted to continue working until they realized that the hardship of their condition precluded them from continuing at the same pace, and they were forced to consider how their work situation could be modified.

#### 3.4.2. Support from the Healthcare Team and the Worksite (System-Related and Worksite VR Factor)

This sub-theme focuses on how healthcare teams and worksites supported participants and how disclosing their cancer diagnosis affected their work. Fourteen participants reported that their healthcare team did not address how to continue working while undergoing cancer treatment, leaving them unheard. However, seven participants who received cancer rehabilitation as part of their cancer survivorship programs found the services beneficial in getting "back in shape for work" (P3). Three participants were unable to find service providers who offered free physical and cognitive assessments. Seven participants who requested psychological support for their cancer-related anxiety and its impact on their work performance were unable to find the assistance they required.

On the other hand, participants who were part of a family-owned business reported that it was easier to manage their work schedule during their cancer treatment. Family-owned businesses frequently assumed some or all of the participants' work responsibilities to allow them time for treatment. The four participants who reported this experience found this assistance especially helpful during their cancer recovery phase: "They left the decision to return to work entirely up to me" (P9). However, ownership in a family-owned business was not always advantageous. One participant explained, "After my hair had grown back, I was asked at family gatherings when I would resume my company duties considering I was 'looking good'. (P18). This pressure to return to work despite still feeling the effects of cancer, such as fatigue and impaired concentration, was reported by a significant number of participants in this study.

Participants reported receiving support from clients when they disclosed their cancer diagnosis and explained why they were unable to perform as much work as before or required assistance with a device. Disclosing their diagnosis made it possible for them to make changes at work without having to conceal or explain them. However, some participants were hesitant to disclose their diagnosis due to the fear of negative impacts on their business or backlash, such as loss of clients or contracts. Despite facing challenges

in physical work ability due to developing osteoporosis and arthritis from chemotherapy, one participant used a cane on the worksite but attributed it to arthritis, choosing not to disclose her condition to clients for fear of retribution to her business. Overall, the extent of client support varied based on communication and disclosure.

3.4.3. Strategies Employed by Self-Employed Individuals with Cancer to Continue and Return to Work (System-Related VR Factor)

In Figure 1, sub-theme 4.3, participants described various strategies they employed to manage their cancer treatment and work responsibilities, often with limited support from healthcare providers. Although the participants acknowledged the high-quality medical care provided by healthcare providers, they felt that they were not adequately prepared to cope with the potential impact of cancer on their ability to remain at work and were the ones to initiate discussions about managing work during treatment. Participants appreciated healthcare providers who tried to accommodate their treatment schedule to their work schedule. For example, some participants asked if it was possible to receive their treatment on Friday afternoons, giving them two days to recuperate before returning to work on Monday. Others found it helpful to have all their treatments booked on the same day every week to plan their workload and work schedule around their treatment schedule. However, some participants had more difficult experiences, with one describing their treatment schedule as an "absolute nightmare" that negatively impacted their ability to maintain their business (P23). Support groups, business mentors, health navigation specialists, and adaptive aids helped participants manage the challenges of continuing to work during and after cancer treatment. Participants incorporated workplace modifications, task reassignments, and temporary staffing to maintain business operations. Technology, such as video calls and recording interactions with clients, was also utilized to assist those whose cancer had physically or cognitively impaired them. Participants showed remarkable resilience and adaptability, using various resources and strategies to manage cancer treatment and continue working.

## 4. Discussion

The findings of this study fill a significant gap in the literature regarding the impact of cancer on self-employed individuals, a particularly under-researched group [31], and even less for studies conducted in Canada [8,14]. This study provides crucial insights into the unique challenges faced by this vulnerable group of workers. As noted in the literature, self-employment is already a precarious state of employment, lacking the job security and access to benefits that typically come with traditional employment [13]. When diagnosed with cancer, the financial repercussions and negative impact on quality of life can be especially devastating for this population [8]. All 23 participants in this study experienced negative work status changes that had significant financial repercussions and decreased their overall quality of life, highlighting the precarious nature of self-employed individuals with cancer. Notable is the fact that one participant turned to GoFundMe to support herself due to the negative impact of late effects of cancer on her work ability (including physical, cognitive, and emotional limitations), which ultimately rendered her business unstainable. In addition, three other participants were faced with taking unplanned early retirement, highlighting the additional challenges of this particular group of workers.

The VR model for cancer survivors was helpful in understanding the specific barriers and facilitators that facilitated the study participants, self-employed individuals with cancer, in staying at work, returning to work and maintaining their businesses [25]. By examining the sub-themes through the lens of the VR model's four factors, the study found that these factors could act as both barriers and facilitators for individuals with cancer returning to work. Biopsychological factors, such as treatment side effects and mental health issues, emerged as significant barriers to re-entry into the workforce, productivity, and function. Personal factors, such as views of the meaning of work, and sociodemographic factors, such as age, played a part in the self-employed individual's decision to continue or return to

work, and for three participants, led to their decision to retire early. System-related factors, such as the lack of access to healthcare, rehabilitation, and financial support, emerged as obstacles to work re-entry and maintenance, as well as balancing work and cancer. The worksite-related factors, including job context, job demands, and workplace relationships, were also identified as critical factors affecting work re-entry and business sustainability during the active phase of the self-employed individual's cancer experience. Overall, the VR model provided a comprehensive framework for understanding the multifaceted factors that impact the work outcomes of self-employed individuals with cancer, thereby guiding the development of interventions to support this population.

The impact of late effects of cancer on work ability and the buffering role of job resources for self-employed individuals with cancer are areas that have been largely unexplored in the literature [32]. This study addresses this gap by identifying the strategies used by 23 self-employed individuals with cancer to maintain their businesses and continue working while managing their cancer. Our findings align with a previous study by Torp et al. [14], which identified specific job characteristics that helped self-employed individuals with cancer retain their work and businesses. Specifically, our findings indicate that job autonomy, flexible work schedules, and workload control played a crucial role in enabling participants to manage their cancer-related late- and long-term side effects while maintaining their business. In addition, some participants dealt with physical, cognitive, and emotional challenges caused by cancer by delegating demanding tasks to employees or contracting out services on a temporary basis in order to continue their business operations. However, these strategies also came with financial and emotional costs, as participants had to bear additional expenses incurred from hiring employees or contracting out services and expressed concerns about the long-term sustainability of their business.

Nevertheless, despite expressing a sense of responsibility to maintain their businesses and support their employees, which likely exacerbates their stress and burden from their cancer [14], job resources and external support systems can play a critical role in reducing this burden [32]. In our study, participants who received support from their healthcare team or family members were less likely to experience business closure and financial difficulties. Informing self-employed individuals about the high likelihood of experiencing cancer and treatment side effects that could affect their work ability could also be beneficial. All but two participants in our study received this information, and all acknowledged not being asked if they needed to remain working after their cancer diagnosis. Having this knowledge in advance could allow self-employed workers to plan contingency measures in the event they experience debilitating side effects from their cancer and treatment that impact their work ability. Job resources, external support systems [32], and having access to professional cancer rehabilitation services [33] can all play a critical role in helping self-employed individuals with cancer maintain their businesses and work ability. Additionally, production-securing measures are required to prevent bankruptcy and support the self-employed with cancer in maintaining their businesses [14]. The first year after a cancer diagnosis is crucial, and having access to financial support during this period is essential.

To provide self-employed individuals with temporary financial support during the critical first year after a cancer diagnosis, similar to the Canadian caregiver program [34], automatic enrollment in employment insurance for 15 weeks could be considered. This would help self-employed individuals maintain their businesses and support their families during times of illness. Additionally, self-employed workers should still have the option to choose the more extended employment insurance plan for self-employed workers offered by the Canadian government if it fits their needs and budget. This plan provides up to 55% of their earnings, up to a maximum of CAD 650 per week for 26 weeks, which may be more beneficial for those requiring longer-term financial assistance.

In certain countries, such as The Netherlands [35] and Sweden, self-employed individuals are automatically enrolled in sickness insurance and have access to sick leave benefits. Other countries, including the United Kingdom [36], have recently implemented sick pay programs for self-employed workers on a voluntary basis. However, many countries,

including the United States [37] and Canada [38], do not provide automatic sick leave coverage for self-employed workers. They are required to purchase private disability insurance or opt for government-run programs.

The provision of short-term and long-term financial support options for self-employed individuals during times of illness could increase their flexibility and control over their financial situation, resulting in improved health outcomes and a more stable economy [39,40]. It is crucial to recognize the financial challenges faced by self-employed individuals with cancer and implement measures to assist them in managing their cancer and business. Exploring alternative options for financial support and developing strategies to assist self-employed individuals with cancer should be a priority of policy makers to ensure their long-term success.

### 4.1. Future Research

Future research is needed to further investigate the impact of cancer on the financial, social, and emotional well-being of the self-employed. Although this study did not specifically address the financial burden of cancer, many participants reported that their diagnosis resulted in the temporary or permanent closure of their business and noted financial losses. To address this issue, our research team will be conducting a study this year that explores the impact of cancer on the financial well-being of the individual and their business, as well as the overall quality of life of the self-employed. In addition, the research can provide recommendations for government labor policies that would support self-employed cancer patients during the active phase of their illness, from diagnosis through the end of treatment. Another area for future investigation is the question of the effectiveness of vocational rehabilitation and occupational programs in mitigating functional impairments, preventing disability, and reducing the impact of cancer on the work capacity and occupational well-being of the self-employed. Such research may offer effective solutions to help self-employed individuals with cancer continue to work and maintain their businesses.

### 4.2. Study Limitations

There are study limitations that must be acknowledged despite the strength of what can be considered a large sample size with 23 self-employed individuals with cancer from diverse job types and regions. The use of self-reported data may be subject to bias or inaccuracies, and response bias may have occurred. In addition, the absence of a control group of self-employed individuals without cancer hinders our ability to determine the extent to which cancer specifically affected work and business outcomes. Moreover, our snowballing and convenience sampling methods may have resulted in an underrepresentation of self-employed individuals from specific areas such as remote and rural, who may face unique difficulties in adapting their work to accommodate cancer treatment. Nevertheless, the heterogeneity and specificity of our sample increase the applicability of our findings to the self-employed Canadian population with cancer.

## 5. Conclusions

In conclusion, this study sheds light on the challenges faced by self-employed individuals with cancer, an under-researched group, particularly in Canada, who are vulnerable to poor work outcomes as a result of the adverse effects of cancer on their physical, cognitive, emotional, social, and financial functions. Our findings highlight the strategies employed by these individuals to overcome these challenges but also underscore the need for supportive policies and interventions tailored to their specific needs. Ultimately, such policies and interventions can lead to improved professional and business outcomes for self-employed individuals with cancer.

**Supplementary Materials:** The following supporting information can be downloaded at: https://www.mdpi.com/article/10.3390/curroncol30050347/s1, Video S1: Testimonial from Frédéric, a Cancer Survivor https://youtu.be/-g_5F9frOQc; File S1: This qualitative research includes a completed report of the 32-item checklist COREQ (Consolidated Criteria for Reporting Qualitative Research) Checklist, which provides a comprehensive framework for reporting qualitative research studies.

**Author Contributions:** Conceptualization, C.M. and M.P.; methodology, C.M.; software, C.M.; validation, C.M., M.P, C.W. and F.Y.; formal analysis, C.M., C.M., M.P., C.W., F.Y. and C.L.; investigation, C.M., C.M., M.P., C.W., F.Y. and C.L.; resources, C.M.; data curation, C.M., C.M., M.P., C.W., F.Y. and C.L.; writing—original draft preparation, C.M., C.W. and F.Y.; writing—review and editing, C.M., M.P., C.W., F.Y. and C.L.; visualization, C.M.; supervision, C.M. and M.P.; project administration, C.M. and C.L.; funding acquisition, C.M. All authors have read and agreed to the published version of the manuscript.

**Funding:** The corresponding author, Christine Maheu, and co-author, Maureen Parkinson, received in-kind support from Cancer and Work (www.cancerandwork.ca). Two of the co-authors, who were both master's students at McGill University's Ingram School of Nursing, received government funding for their collaboration on this study. Specifically, Caitlin Wong was supported by a Canada Graduate Scholarship Master's Award from the Canadian Institutes of Health Research (1 April 2021), while Fatima Yashmin was supported by the Ministry of Education and Graduate Studies (MEES-Q) and Nursing Orders of Quebec (OIIQ) (20 August 2020). We are grateful to Réseau de recherche en interventions en sciences infirmières du Québec/Quebec Network on Nursing Interventions Research (RRISIQ) for their generous grant support, which has enabled us to promote open access and disseminate our research widely (30 March 2023).

**Institutional Review Board Statement:** The study was conducted in accordance with the Declaration of Helsinki, and approval by the Institutional Review Board of the McGill Faculty of Medicine and Health Sciences (McGill University Health Centre (MUHC) #MM-2021-6784, date of approval: 21 August 2021) was granted to the corresponding author, C. Maheu.

**Informed Consent Statement:** Informed consent was obtained from all subjects involved in the study.

**Data Availability Statement:** The datasets are unavailable online due to privacy or ethical restrictions. Data are available from the corresponding author upon reasonable request.

**Acknowledgments:** This study is indebted to the 23 self-employed individuals who shared their experiences, providing valuable insights into the unique challenges self-employed individuals face to maintain their businesses and continue working while managing their cancer.

**Conflicts of Interest:** The authors have no conflict of interest. The funders played no role in the study design, data collection, analysis, manuscript preparation, or publication decision.

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
