# Peer review of "Self-Employed Canadians’ Experiences with Cancer and Work: A Qualitative Study"

_curroncol, doi:10.3390/curroncol30050347_

Round 1

Reviewer 1 Report

Thank you for the opportunity to review this article. The article is well written and provides an understanding of experiences of self employed people who are receiving cancer treatment. Please see comments for each section below:

Introduction

At the end of the introduction please provide an aim followed by the research questions. Currently there is an incomplete sentence at the beginning of the paragraph with research questions.

Methods

The methods section does not flow. It would be good to use subheadings and make sure the paragraphs flow. For example the first paragraph in the methods starts with qualitative design then includes recruitment and sampling and ethics.

Consider using the COREQ guidelines or describe steps to ensure rigour of the study.

Results

Were the demographic details collected in a survey – this is not clear? Might help to explain why 4 participants did not answer the question on self employed. Were these participants definitely self employed?

Please clarify for the participant who was retired – was this only after their cancer diagnosis? 

Figure 1 is great

The themes are detailed and presented well with exemplars.

Discussion

The summary at the end of the results might fit better in the discussion/ it seems to overlap with the first paragraph of the discussion – please check for overlap and repetition.

Well written discussion overall

Ideas for future research include evaluating a vocational rehab program – it would be good to reference previous studies that have tested such programs in other countries and or in Canada?

Please provide a limitations section.

Author Response

Reviewer 1

Comment 1 Thank you for the opportunity to review this article. The article is well written and provides an understanding of experiences of self-employed people who are receiving cancer treatment. Please see comments for each section below:

Thank you for your review of our manuscript. We appreciate your gracious comments and are pleased to learn that our manuscript provides valuable insight into the impact of cancer on self-employed individuals. We have carefully considered all of your comments and have addressed them diligently. We hope that these revisions have improved the clarity and structure of our manuscript. Thank you for your time and valuable feedback.

Comment 2: Introduction

At the end of the introduction please provide an aim followed by the research questions. Currently there is an incomplete sentence at the beginning of the paragraph with research questions.

Thank you for your feedback and for highlighting the incomplete sentence in the introduction. We have addressed this issue and provided a clear objective and research questions at the conclusion of the introductory section.

This qualitative interpretive description study aimed to explore the impact of cancer on self-employed Canadians, including their health, work ability, and business, as well as identify the challenges they faced in maintaining their work and business during and after their cancer diagnosis. Additionally, the study sought to investigate the strategies used by self-employed Canadians diagnosed with cancer to continue working and maintain their business while managing their cancer. Finally, the study aimed to suggest potential support programs or policies that could better meet the needs of this population.

Comment 3: Methods

The methods section does not flow. It would be good to use subheadings and make sure the paragraphs flow. For example, the first paragraph in the methods starts with the qualitative design and then includes type of sampling (how the participants were selected, such as convenience, snowball) and description of sample (inclusion and exclusion criteria), recruitment (how participants were approached), sample size, setting (where the data was collected), data collection (interview guide, how the guide was produced, what were examples of some of the question, what were the recording devise use, how long were the interviews, was data saturation discussed, transcription and translation done and by whom), data analysis ( how many coders coded the data, was a description of the coding scheme provided, were the themes identified in advanced or derived from the data, what software was used if any), reporting ( were participants quotations presented to illustrate themes/findings? Was each quotation identified with participant number?

Consider using the COREQ guidelines or describe steps to ensure rigour of the study.

We appreciate your comprehensive review of our manuscript. We appreciate your suggestion to use the COREQ guidelines to improve the clarity of the methods section and increase the rigour of our study.  Using the COREQ guidelines, we have revised and reorganized the section to ensure that the procedures and analyses are reported clearly and exhaustively. In addition, we have completed the 32-item COREQ checklist, and identified the corresponding page number where each item was addressed in the manuscript.

We thank the reviewer for their insightful comments, which prompted us to clarify the association between participant quotations and unique participant numbers in our manuscript. Specifically, we added the following sentence after describing Table 1 and how the presentations of participants’ quotes represent raw data. 'To ensure traceability and identification of every participant's quotations, each quote was assigned a unique participant number, which was used throughout the analysis process.'

We hope this addition addresses the reviewer's concern and provides further transparency in our data analysis. Thank you for your valuable feedback.

Comment 4: Results

Were the demographic details collected in a survey – this is not clear? Might help to explain why 4 participants did not answer the question on self employed. Were these participants definitely self employed? Please clarify for the participant who was retired – was this only after their cancer diagnosis? Figure 1 is great. The themes are detailed and presented well with exemplars.

Thank you for bringing to our attention the oversight of not providing information on how the demographic data was collected. We apologize for this oversight and any confusion it may have caused.

We have added to our manuscript how we collected the demographic data through a personalized link to Qualtrics, a secure web-based electronic data capture system. Participants were asked to complete the demographic questionnaire after providing their consent by checking the consent box directly on the electronic form.

Comment 5: Are all participants self-employed and why 4 participants did not answer a question on the demographic questionnaire

All participants included in our study met the inclusion criterion of being self-employed individuals. Regarding the four participants who did not answer the question about being self-employed, it is important to note that they did not respond to the years in business question, and not to whether they were self-employed or not. Additionally, we would like to clarify that the participant who is listed as retired in Table 1 did so after their cancer diagnosis. We apologize for any confusion this may have caused and have updated the information in Table 1 accordingly. We hope this clarifies any ambiguity and addresses the concerns of the reviewers. Thank you for your insightful comments.

Comment 6: Discussion

The summary at the end of the results might fit better in the discussion/ it seems to overlap with the first paragraph of the discussion – please check for overlap and repetition. Well written discussion overall.

We appreciate your insightful comments. As suggested, we have revised the manuscript to include a concise summary of the themes and subthemes identified in the study at the beginning of the discussion section only. We have ensured that there is no duplication or repetition in the discussions from the text discussed in the results section. Thank you for helping us improve our manuscript's organization and clarity.

Comment 6: Ideas for future research include evaluating a vocational rehab program – it would be good to reference previous studies that have tested such programs in other countries and or in Canada? And Comment 7) Please provide a limitations section.

Thank you for suggesting that a section on future research and study limitations be included. We have taken into account your feedback and added sections for both subjects just before the conclusion under 5.1 Future Research and 5.2 Study Limitations.

In the Future Research section, we acknowledge the need for further investigation into the impact of cancer on the financial, social, and emotional well-being of the self-employed. In addition, we propose investigating the efficacy of vocational rehabilitation and occupational programs in mitigating functional impairments and minimizing the impact of cancer on work capacity and occupational well-being. In addition, we propose that future research could provide recommendations for government labour policies that support self-employed cancer patients during their cancer recovery phase. We appreciate your constructive suggestions for improving the manuscript.

Reviewer 2 Report

It's an honor to review your great research. This is a necessary research on an important part of cancer survivors' return to life.

A few review comments.

1. Research methods: All forms of qualitative research are inductive approaches.

Your research borrowed interview questions from the VR model to find answers to your research questions. This is not appropriate as an interview question in qualitative research.

Interview questions developed by researchers for data collection are valid tools.

Please state the question.

2. In the analysis of research results, it was described that the initial analysis was thematic analysis, and the content analysis was applied afterwards.

By the way, the research results were organized according to the factors of the VR model.

This goes against the basic framework of inductive qualitative research. The main word of the result must be presented as an inductive analysis result.

3. Please present a table of results of subject analysis and content analysis of research result analysis.

4. Afterwards, it is recommended to conduct a secondary analysis according to the VR factor.

5. The results section takes up too much space.

  It seems necessary to abbreviate each topic to be newly analyzed.

6. Table 2 may be added to Table 1.

7. For a global reader, residence is not thought to be a meaningful characteristic, but rather, gender is curious. The type of self-employment is thought to differ by gender.

8. The discussion should not redundantly describe the contents of the results.

Modification of the result analysis part is required.

Author Response

Reviewer #2

Comment 1, 2, and 3: few review comments.

Comment 1: Research methods: All forms of qualitative research are inductive approaches. Your research borrowed interview questions from the VR model to find answers to your research questions. This is not appropriate as an interview question in qualitative research. Interview questions developed by researchers for data collection are valid tools. Please state the question.

Comment 2: In the analysis of research results, it was described that the initial analysis was thematic analysis, and the content analysis was applied afterwards.

Comment 3; By the way, the research results were organized according to the factors of the VR model. This goes against the basic framework of inductive qualitative research. The main work of the result must be presented as an inductive analysis result. Afterwards, it is recommended to conduct a secondary analysis according to the VR factor.

We would like to thank the reviewer for their thorough examination of our manuscript and insightful comments that helped us improve the clarity and transparency of our research methods and findings. We appreciate the opportunity to respond to each of the reviewer's comments in this response. We will specifically address the reviewer's concerns regarding the appropriateness of borrowing interview questions from the VR model, the order of analysis methods used in our study, and the organization of our research results in accordance with the VR model factors. We hope that our responses provide the necessary clarifications and address the reviewer's concerns to their satisfaction.

In response to your first comment, we have revised our manuscript to better articulate our research question. We have included a statement that outlines the aim of our qualitative interpretive description study, which is to explore the impact of cancer on self-employed Canadians, including their health, work ability, and business, as well as identify the challenges they face in maintaining their work and business after cancer. Additionally, we aimed to investigate the strategies used by self-employed Canadians diagnosed with cancer to continue working and maintain their business while managing their cancer. Finally, we aimed to suggest potential support programs or policies that could better meet the needs of this population.

Regarding the reviewer’s concern about using the VR model in our study, we would like to assure that our analysis was conducted inductively, with the data guiding the generation of themes and sub-themes. The VR model was not imposed on the data; rather, it was used as a framework to understand the various factors that contribute to returning to work after cancer treatment and to identify specific challenges and facilitators within each factor. After all the codes were grouped and linked to specific sub-themes and themes, we then analyzed the sub-themes by linking them to the four-factor VR model for cancer survivors. This approach allowed us to examine comprehensively how cancer affected the health, work ability, and business of self-employed individuals, to identify facilitators and barriers with returning to work, maintaining work, and their business, and to help explain work-related challenges experienced by self-employed individuals following cancer. It also helped to provide insights regarding potential components to consider in developing vocational rehabilitation and occupational programs for self-employed individuals with cancer to mitigate functional impairments and minimize the impact of cancer on work capacity and occupational well-being.

Under our data analysis section, we have added more details to describe how we analyzed our data inductively and linked the findings to the VR model only after all codes were analyzed. We have included a detailed step-by-step description of our six-phase analytical process, including our use of in vivo coding, creation of sub-themes and candidate themes, recursive review, and thematic framework development. We have also included a table (Table 1) that provides specific examples of how we linked the participants' raw data to specific sub-themes and factors in the VR model.

We believe that our analytical approach, which utilized the VR model to enhance our analysis, provides a transparent and rigorous methodology for investigating the impact of cancer on the health, work ability, and business of self-employed Canadians. We hope this clarification addresses the concerns raised by the reviewer.

Comment 4 and 5: Please present a table of results of subject analysis and content analysis of research result analysis.

Thank you for your valuable feedback. We appreciate your suggestion to present a table illustrating the progression from content to subthemes and themes. We have created Table 1, which details participant quotes in relation to codes, subthemes, and themes. We hope this table contributes to the readability of our manuscript by illustrating our data analysis strategy and results presentation.

Comment 6: The results section takes up too much space. It seems necessary to abbreviate each topic to be newly analyzed.

We appreciate the reviewer's feedback and have carefully considered their comments. In response to the concern regarding the length of the results section, we have synthesized and reduced it by half by abbreviating each theme and subtheme that was analyzed. We believe this new format enhances the clarity and readability of the results section while providing a comprehensive summary of the findings. We hope that this modification addresses the reviewer's concerns.

Comment 7: Table 2 may be added to Table 1.

Thank you for your recommendation about Table 2. In response to your suggestion, we have merged the two tables into Table 2, which provides a comprehensive overview of the demographic, clinical, and work-related characteristics of the participants, as well as their work status following a cancer diagnosis, using the Canadian National Occupational Classification (NOC). In addition, in response to another reviewer's comment, we have created a new Table 1 that describes the transformation of participant quotes from raw data to codes, subthemes, and themes. We hope that these modifications address your comment and provide a clear and thorough representation of the findings of our study.

Comment 8: For a global reader, residence is not thought to be a meaningful characteristic, but rather, gender is curious. The type of self-employment is thought to differ by gender.

Thank you for your feedback regarding the residence of our Canadian participants. We acknowledge that the term "residence" may not be meaningful to all readers and apologize for any confusion this may have caused. As a result, this characteristic has been modified to indicate the Canadian province. The term "province" refers to the specific Canadian region in which the participants reside or are based. This information is pertinent to the study because it contextualizes the location of the participants and their potential access to resources and support systems based on their geographic location. We believe this modification will provide more specific information about the location of the participants without causing additional confusion. Thank you again for your helpful comment.

Comment 9: The discussion should not redundantly describe the contents of the results. Modification of the result analysis part is required.

Thank you for your comments about the discussion section. We have carefully reviewed the discussion and eliminated duplication with the results section while emphasizing the significance of the key findings. We also added defining subsections to the discussion, where we discussed future research recommendations and study limitations. To address the reviewer's concerns regarding the modification of the result analysis, we also elaborated on our analysis methodology. We believe these modifications have improved the clarity and structure of the discussion section, and the whole manuscript. We value the reviewer's comments and suggestions for enhancing the quality of our manuscript.

Reviewer 3 Report

This manuscript addresses an important issue relating to a very under-researched population of self-employed cancer survivors. The article is well written and the methodology was thorough and appropriate for capturing their experiences. I read with great interest and feel it will be well received by Current Oncology readers given the rising numbers of self-employed Canadians facing cancer. I recommend this for publication in Current Oncology.

Author Response

Reviewer 3

Comment 1: General comment

This manuscript addresses an important issue relating to a very under-researched population of self-employed cancer survivors. The article is well written and the methodology was thorough and appropriate for capturing their experiences. I read with great interest and feel it will be well received by Current Oncology readers given the rising numbers of self-employed Canadians facing cancer. I recommend this for publication in Current Oncology.

Thank you for reviewing our manuscript and for your constructive feedback. We are pleased you found our study informative and well-written and that our method for capturing self-employed cancer survivors' experiences was appropriate. We appreciate your suggestion for the Current Oncology publication and hope our research will make a significant contribution to the field.

Round 2

Reviewer 2 Report

It took a lot of effort to revise many parts of the article. It's been cleaned up a lot.

The setting of 2.3 should be divided into 2.2 Participant Selection and Method of Approach and 2.4 Data Collection. The setting session is separate which further confuses the reader.

Author Response

Response 1: Dear Reviewer 2,

Thank you for taking to give a second reading and review our manuscript and for providing us with valuable feedback. We appreciate your insightful comment regarding the setting of Section 2.3 and agree that it could be confusing for readers. As such, we have made the changes you suggested by dividing the section into two subsections, 2.2 Participant Selection and Method of Approach and 2.3 Data Collection, with 2.4 Data Analysis as the final subsection (page 4, line 168).

We combined the two paragraphs that started with "All potential participants were invited to participate" and "After consenting and completing the questionnaires" with the Participant Selection and Method of Approach section. The remaining two paragraphs were added to Data Collection. We believe that these changes have resulted in a clearer and more logical organization of the section.

I have made one minor change in the Results section, page 6, line 244, from "The majority of participants" to "Most participants" to enhance clarity. 

Thank you again for your insightful feedback, which has helped us improve the manuscript. We hope these revisions have addressed your concerns and improved the clarity of our manuscript. Thank you for your helpful feedback.

Best regards,

Christine Maheu